# *APC* and *TP53* Mutations Predict Cetuximab Sensitivity across Consensus Molecular Subtypes

**DOI:** 10.3390/cancers13215394

**Published:** 2021-10-27

**Authors:** Ramya Thota, Mingli Yang, Lance Pflieger, Michael J. Schell, Malini Rajan, Thomas B. Davis, Heiman Wang, Angela Presson, Warren Jack Pledger, Timothy J. Yeatman

**Affiliations:** 1Oncology Clinical Program, Intermountain Healthcare, Murray, UT 84107, USA; 2Department of Surgery, University of Utah, Salt Lake City, UT 84132, USA; mingli.yang@utah.edu (M.Y.); mrajan@u2m2.utah.edu (M.R.); thomas.davis@utah.edu (T.B.D.); heiman.wang@utah.edu (H.W.); jack.pledger@utah.edu (W.J.P.); 3Precision Genomics Translational Science Center, Intermountain Healthcare, Murray, UT 84107, USA; Lance.Pflieger@imail.org; 4Department of Biostatistics and Bioinformatics, Moffitt Cancer Center & Research Institute, Tampa, FL 33612, USA; Michael.Schell@moffitt.org; 5Department of Internal Medicine, Division of Epidemiology, University of Utah, Salt Lake City, UT 84132, USA; angela.presson@hsc.utah.edu; 6Huntsman Cancer Institute, University of Utah, Salt Lake City, UT 84112, USA

**Keywords:** colorectal cancer, *APC*, *TP53*, mutations, CMS classification, EGFR inhibitors, cetuximab

## Abstract

**Simple Summary:**

Colorectal cancer (CRC) is a major cause of cancer deaths. Cetuximab is an FDA-approved, underutilized therapeutic targeting the epidermal growth factor receptor (EGFR) in metastatic CRC. To date, despite selection of patients with wild-type RAS, it is still difficult to identify patients who may benefit from EGFR inhibitor (e.g., cetuximab) therapy. Our aim is to molecularly classify CRC patients to better identify subpopulations sensitive to EGFR targeted therapy. *APC* and *TP53* are two major tumor suppressor genes in CRC whose mutations contribute to tumor initiation and progression and may identify cetuximab-sensitive tumors. Recently, it has been suggested that the consensus molecular subtype (CMS) classification may be used to help identify cetuximab-sensitive patients. Here, we report an analysis of multiple CRC tumor/PDX/cell line datasets using combined *APC* and *TP53* mutations to refine the CMS classification to better predict responses to cetuximab to improve patient outcomes.

**Abstract:**

Recently, it was suggested that consensus molecular subtyping (CMS) may aide in predicting response to EGFR inhibitor (cetuximab) therapies. We recently identified that *APC* and *TP53* as two tumor suppressor genes, when mutated, may enhance cetuximab sensitivity and may represent easily measured biomarkers in tumors or blood. Our study aimed to use *APC* and *TP53* mutations (AP) to refine the CMS classification to better predict responses to cetuximab. In total, 433 CRC tumors were classified into CMS1-4 subtypes. The cetuximab sensitivity (CTX-S) signature scores of AP vs. non-AP tumors were determined across each of the CMS classes. Tumors harboring combined AP mutations were predominantly enriched in the CMS2 class, and to a lesser degree, in the CMS4 class. On the other hand, AP mutated CRCs had significantly higher CTX-S scores compared to non-AP CRCs across *all* CMS classes. Similar results were also obtained in independent TCGA tumor collections (*n* = 531) and in PDMR PDX/PDO/PDC models (*n* = 477). In addition, the in vitro cetuximab growth inhibition was preferentially associated with the CMS2 cell lines harboring A/P genotypes. In conclusion, the AP mutation signature represents a convenient biomarker that refines the CMS classification to identify CRC subpopulations predicted to be sensitive to EGFR targeted therapies.

## 1. Introduction

Epidermal Growth Factor Receptor (EGFR) inhibitors are effective in the subset of RAS (*KRAS, NRAS*) wild-type colorectal cancer (CRC) patients. Even among RAS wild-type patients, who account for approximately 40 percent of all metastatic CRC [1], only approximately 50–60 percent of patients derive benefit from these drugs, suggesting that additional genes—beyond RAS—may negatively contribute to EGFR inhibitor (EGFRi) response [2,3,4,5]. Beyond RAS mutation, primary tumor sidedness (right-sided or left-sided) also predicts response to EGFR inhibitors, with right-sided tumors having worse survival with EGFR inhibitor therapy [6,7]. Recently, a few studies have reported a conflicting, but possibly predictive role, of consensus molecular subtype (CMS) classification for the selection of patients for EGFRi therapy [8,9].

The CMS classification is one of the most comprehensive molecular analyses of human tumors developed to best define the principal components of CRC [10]. The classification categorizes CRC into one of four consensus molecular subtypes with distinguishing features: CMS1 (immune), hypermutated, microsatellite unstable tumors with strong immune activation; CMS2 (canonical), epithelial, with marked WNT and MYC signaling activation; CMS3 (metabolic), epithelial and with evident metabolic dysregulation; CMS4 (mesenchymal), characterized by transforming growth factor–β activation, stromal invasion and angiogenesis, although a fair number remain unclassified. The prevalence of CMS classes varies by tumor sidedness and stage of the tumors with CMS1 and CMS3 subtypes enriched in right-sided tumors and early-stage CRC, while CMS2 enrichment was noted in left-sided tumors and CMS4 class most common in advanced tumors [10,11,12,13,14].

The prognostic and predictive factors of CMS classes were demonstrated in retrospective analysis of multiple phase III clinical trials [8,9,14,15,16,17,18,19]. The prognostic role of CMS classification was well established in most of the studies, with prognosis of each CMS class varying from early-stage to advanced-stage tumors [8,9,10,14,15,20]. While the CMS2 class has the best prognosis overall, the CMS1 class has the best prognosis in early stage cancers, but they harbor very poor outcomes in advanced stage CRC [8,16]. Ultimately, it was felt the prognosis is informed by the interaction of the genomic markers such as RAS, *BRAF*, and MSI with CMS and the tumor microenvironment [21,22].

Despite the established *prognostic* role, there remains conflicting data on the *predictive* role of CMS classification especially, in the metastatic setting. For instance, in early-stage tumors, CMS2 enrichment was suggested to derive benefit from oxaliplatin in stage II-III tumors in the CO-7 study but the results from the MOSAIC study do not support these findings [17,18]. In the metastatic setting, CMS2, and CMS4 classes were noted to have better outcomes with EGFR inhibitor therapy in two separate Phase III studies [8,9]. Likewise, CMS2 and CMS3 classes were noted to have better outcomes with anti-vascular endothelial growth factor (VEGF) therapy [14,20]. The conflicting predictive role of CMS classification was felt due to multiple reasons including complex genomic interplay between number of the genomic markers beyond RAS such as *BRAF* and/or hypermutated tumors that would confer innate resistance to EGFRi therapy in these tumors [12,21]. The conflicting role suggested a need to refine this classification system.

In order to better refine the CMS classification, the combination of CMS with other signatures such as the recombinant proficiency score (RPS) or DNA damage repair score, were studied to determine if patients with non-stem-like tumors and low RPS scores (i.e., non-CMS4 patients) would achieve significant benefit with oxaliplatin [17]. In the future, integration of the genomic pathway signature scores with transcriptomic analysis may help develop novel biomarkers that predict drug sensitivity.

*APC* and *TP53* are the major tumor suppressor genes which are frequently mutated in CRC and play a key role in tumor initiation and progression [23,24,25,26,27]. We previously reported the prognostic role for *APC* that relates to the number of alleles mutated and to the association with other mutant genes such as *KRAS* and *TP53* using a database of 468 molecularly profiled CRCs including global gene expression and targeted exome sequencing [27]. Further analysis also revealed that mutant *APC* genotypes, in combination with mutant *TP53*, strongly correlated with a gene expression signature measuring cetuximab sensitivity (CTX-S), suggesting a predictive role of the 2-gene mutation signature of *APC* and *TP53* mutations (AP) in CRC [28]. In this study, we aimed to understand the best predictors of response to EGFRi therapy beyond the consensus molecular subtypes, in an attempt to best select CRC patients that might derive maximal benefit from EGFRi. Specifically, we investigated the potential for AP mutations to refine the predictive value of the CMS classification for cetuximab sensitivity.

## 2. Materials and Methods

### 2.1. Moffitt CRC Patient Samples

We retrospectively analyzed 468 tumor samples with stage I–IV colorectal adenocarcinoma patients accrued between October 2006 and September 2010 as part of the Total Cancer Care (TCC) project through a collaboration between Merck and Moffitt Cancer Center as reported previously [27,29]. All of the experimental protocols involving human data were performed in accordance with the guidelines of national/international/institutional or Declaration of Helsinki. This study was approved by the Institutional Review Board of Moffitt Cancer Center as part of the Total Cancer Care^®^ (TCC) project (MCC14690) and written informed consent was obtained from all the participants [30]. The tumor samples were collected at the time of curative surgery and immediately frozen within 15–20 min of extirpation. The samples then were analyzed for quality control to ensure > 80% of the tumor present in the sample. Subsequently, DNA was extracted and submitted for targeted exome sequencing (1321 genes) and RNA was extracted from the same sample for Affymetrix global gene expression analysis [27,29].

### 2.2. CMS Classification

The global gene expression profiling of 468 CRC tumors was described in our previous studies [27,28,29]. Using these comprehensive gene expression data, we assigned CMS classification to 468 samples using CMScaller algorithm in combination with the Random Forest classification (RF) and the single sample predictor (SSP) classification methods as previously described [10,27,31]. In total, 433 tumors were classified as CMS1-4 subtypes, whereas 35 tumors were CMS-NA (not applicable to any CMS subtype) that were excluded from further analysis. CMScaller was used with default options.

### 2.3. Cetuximab Sensitivity Signature Score

A prespecified and validated 203-gene expression signature score that measures cetuximab sensitivity (CTX-S) we previously developed [28] was used as a surrogate marker for response to cetuximab in this study. This CTX-S score was based on the gene expression values from >800 cancer-associated genes, each assessed from *KRAS* WT colon tumor samples treated with cetuximab monotherapy. The full details of the gene signature score derivation, validation, and overall methodology are previously reported [28].

### 2.4. Validation CRC Datasets

TCGA CRC tumors (*n* = 531): The Cancer Genome Atlas (TCGA) Pan-Cancer Atlas dataset was used to validate the findings from Moffitt CRC [32]. The Pan-Cancer dataset was created using standardized workflows and consists of over 11,000 tumors from 33 cancers. Information on quality control metrics, alignment, and batch-correction are outlined in the Pan-Cancer publication or at https://gdc.cancer.gov/about-data/publications/pancanatlas (accessed on 27 April 2021). For this analysis, the Colorectal Adenocarcinoma (TCGA, PanCancer Atlas) cBioportal dataset was downloaded from the cBioportal repository (https://www.cbioportal.org/datasets, accessed on 27 April 2021). This dataset includes gene expression values, somatic mutations and corresponding clinical data. The RNA-seq values from cbioportal (RNASeq V2 pipeline) are batch-corrected RSEM median normalized count values derived from the Pan-Cancer Atlas alignment files [33]. Values were log2 normalized for gene signature and CMS classification. In total, 531 TCGA tumors were classified by CMScaller as CMS1-4 subtypes and were used in the AP/CTX-S score analysis across CMS classes. Of note, PCA analysis was performed on the TCGA CRC normalized data for outlier detection and QC (see Appendix A).

NCI PDMR models (*n* = 433): The NCI Patient-Derived Models Repository (PDMR) CRC dataset was used for the validation analysis. The PDMR analysis consists of all adenocarcinoma samples in the PDMR repository (https://pdmr.cancer.gov/, accessed on 23 January 2021). Gene level count values from RSEM were pulled from the PDMR FTP server (ftp://dctdftp.nci.nih.gov/pub/pdm/, accessed on 23 January 2021). Only the latest pipeline version available for each sample was used in the analysis. Count values were log2 median normalized to keep analysis similar to the TCGA dataset and match CMScaller default normalization. Oncokb gene panel data containing mutation status was similarly pulled for each sample. A total of 477 PDMR CRC models (primarily patient-derived xenografts) classified by CMScaller as the CMS1-4 subtypes were used in the AP/CTX-S score analysis across CMS classes. Of note, PCA analysis of PDMR CRC data was performed for QC (see Appendix A).

Medico CRC cell lines (*n* = 91): Recently, expression values for 155 CRC cell lines with heterogeneous genetic backgrounds were reported by Medico et al. for genetic and transcriptional profiling of CRC cells and in vitro cetuximab sensitivity [34]. We previously used these cell line data to validate a cetuximab sensitivity signature score we developed [28]. To further validate the finding from Moffitt CRC tumors, Loess normalized values from the Illumina HumanHT-12 V4.0 expression beadchip (Illumina Inc, San Diego, CA, USA) were downloaded via the R package GEOquery with accession number GSE59857 [35]. Illumina probe IDs were mapped to gene symbols using the R package AnnotationDBI. Multi-mapping probes were collapsed into the mean value for each gene symbol [36] followed by log2 normalization. Mutation status for Medico CRC cell line analysis was downloaded from public knowledge bases, the Cancer Cell Line Encyclopedia (CCLE, https://sites.broadinstitute.org/ccle/, accessed on 19 November 2020) and the Cancer Dependency Map (DepMap, https://depmap.org/portal/, accessed on 19 November 2020). In total, 91 cell lines had both the gene expression and *APC/TP53/KRAS/NRAS/BRAF* data, among which 67 cell lines were classified by CMScaller as the CMS1-4 subtypes. These cell lines were used in the AP/CMS/in vitro CTX sensitivity analysis. Of note, PCA analysis of CRC cell line data was performed for QC (see Appendix A).

Notably, for PCA quality control analysis, principle components were generated using the top 1000 variable genes after normalization (RNAseq data is median log2 normalized, array data is Loess log2 normalized). No outliers were detected.

### 2.5. Statistical Analyses

Baseline characteristics in Moffitt CRC tumors were analyzed. Age was compared across CMS classes using an analysis of variance test (ANOVA) with a Tukey post hoc test. Specimen type, primary tumor location, and mutation types were compared across subtypes using a chi-squared test. We performed Kaplan–Meier (KM) survival analysis in the Moffitt AP vs. non-AP tumors. Two-tailed Welch t-tests were performed for comparison analysis in the Moffitt dataset as well as additional validation CRC datasets. All tests were 2-sided with an unadjusted *p*-value less than 0.05 chosen as significant. Of note, for TCGA and PDMR RNAseq data, the CTX-S signature scores were generated using the ssGSEA methodology. These statistical analyses were performed using GraphPad Prism version 8.00 (GraphPad Software, Inc., La Jolla, CA, USA) and R version 3.6.2 (R Core Team, Vienna, Austria).

## 3. Results

### 3.1. Baseline Characteristics of Moffitt CRC Tumors

The baseline characteristics are listed in Table 1. Among 468 tumor samples analyzed, CMS1-4 subtypes could be determined in 433 samples. The frequencies of CMS 1-4 were as follows: CMS1 (*n* = 74; 17%), CMS2 (*n* = 169, 39%), CMS3 (*n* = 85; 20%), and CMS4 (*n* = 105; 24%). We noted earlier age of onset in CMS2 and later age of onset in CMS1 subtypes (ANOVA *p* < 0.001). Further pairwise comparison shows that significant age differences were seen between CMS1 vs. CMS2 and CMS1 vs. CMS4 (Appendix A). CMS4 subtype has patients with more advanced stage at diagnosis. CMS2 are more enriched with left tumors while right sided tumors were commonly seen in the CMS1 subtype. APC mutations were noted with higher frequency in CMS2 and are least commonly seen in CMS1 subtype. Similarly, TP53 mutations were most common in CMS2 but least common in CMS3 subtype. As expected, we noted enrichment of RAS mutations in CMS3 subtype and lowest frequency in CMS1. BRAF mutations and MSI-high tumors were most common in CMS1 while least common in CMS2. The primary tumors represented 79% and the rest were metastatic samples. The majority of the tumors were early stage (I–III), and about one-fourth of tumors were stage IV (*n* = 100; 23%). We noted that the majority of CMS2 tumors were left sided while CMS1 tumors were right sided. We also assessed the prevalence of combined APC and TP53 mutations (AP) based on tumor sidedness and found that the majority of AP mutant tumors are left sided (71% vs. 28%).

### 3.2. Frequency of 2-Gene AP Mutation Signature across the CMS Classes

Combined APC and TP53 mutations (AP) were noted in 101 patient samples (23%). The high prevalence of AP mutations across CMS subtypes was seen in CMS2 (73%) followed by CMS4 (21%) and CMS3 (5%) with the least common frequency observed in CMS1 (2%) (Figure 1A). The distribution of combined AP mutations in RAS/RAF wild-type and mutant tumors is shown in Figure 1B,C, respectively. Similar patterns were noted with high frequency in CMS2 followed by CMS4, CMS3, and CMS1, respectively. The details of the stage-specific distribution of the AP mutations are noted in Table 2.

### 3.3. Predictive Role of 2-Gene AP Mutation Signature across the CMS Classes

Despite a substantially higher frequency in CMS2 tumors, AP mutant tumors had higher CTX-S scores than nonAP tumors (Figure 2) across all the CMS classes. The CMS2 class had overall high CTX-S scores irrespective of AP mutational status. Among non-CMS2 classes, CTX-S scores were significantly higher in AP mutant tumors compared to nonAP tumors.

In addition, among the RAS/RAF wild type tumors, despite their known sensitivity to EGFR inhibitor therapy, we noted that CMS1 and CMS3 tumors as well as nonAP CMS4 tumors had lower CTX-S scores (see Appendix A), suggesting the likelihood of resistance to EGFR inhibitor therapy.

Interestingly, when RAS (KRAS/NRAS)-mutated CRC tumors were associated with combined AP mutations (i.e., APK mutations), the CTX-S scores were significantly higher (see Appendix A). This was seen not only in the CMS2 class, but also in the other CMS cohorts (Appendix A). These results suggest that combined AP mutations can select a subset of patients within each CMS cohort that may benefit most from EGFR inhibitor therapy, thereby expanding the utility of this drug class.

### 3.4. Prognostic Role of 2-Gene AP Mutation Signature and CMS Classes

There was no difference in OS noted between AP mutant and non-AP mutant tumors (Figure 3A) suggesting a predictive but not prognostic value. The median overall survival (OS) across the CMS subtypes was CMS1—45 months; CMS2—78 months; CMS3—65 months; CMSS4—67 months (*p* = 0.096). Notably, although CMS1 tumors appeared to have lower OS, no statistically significant survival differences were noted across the CMS classes (Figure 3B).

### 3.5. Predictive Role of 2-Gene AP Mutation Signature/CMS Classes in TCGA CRC Tumors

To validate the findings from Moffitt CRC tumors, we analyzed the TCGA dataset that consists of all Colorectal Adenocarcinoma samples from the PanCancer Atlas dataset [32]. In total, 531 TCGA tumors were divided into 6 mutation subgroups based on their mutation status of APC(A), TP53(P), and KRAS(K)—3 MUT KRAS subgroups: (1) APK (*n* = 82); (2) AK or PK (*n* = 89); (3) K (*n* = 15) and 3 WT KRAS subgroups: (4) AP (*n* = 138); (5) A or P (*n* = 79); (6) WT AP (*n* = 128). The CTX-S score comparison analysis among these subgroups (Figure 4) shows that the AP tumors had significantly higher scores than all other subgroups (*p* < 0.0001). Notably, the APK (APC/TP53/KRAS triple-mutated) tumors had the second highest scores that were significantly higher than other subgroups lacking combined APC and TP53 mutations, regardless of KRAS mutation status.

These 531 TCGA tumors were also classified into four CMS classes: CMS1 (*n* = 92); CMS2 (*n* = 162); CMS3 (*n* = 93); CMS4 (*n* = 184). The CMS2 tumors had the highest CTX-S scores that were significantly higher than all other three CMS classes (*p* < 0.0001) (Figure 5A). Moreover, the CMS4 tumors also had higher scores than the CMS1 and CMS3 tumors (*p* < 0.0001) (Figure 5A). Notably, the AP tumors appear much more frequently in CMS2 or CMS4 than CMS1 or CMS3 (Figure 5B). Importantly, the tumors harboring combined APC and TP53 mutations (AP/APK) had significantly higher CTX-S scores than other nonAP/APK tumors in CMS2 or CMS4 classes (Figure 6). Similar results were also seen when comparison was done in 345 WT KRAS and 186 MUT KRAS CMS1-4 CRCs, respectively (see Appendix A).

### 3.6. Predictive Role of 2-Gene AP Mutation Signature/CMS Classes in PDMR CRC Models

The NCI Patient Derived Models Repository (PDMR)) contained ~500 CRC patient-derived xenograft (PDX) models as well as their derived organoids (PDOs) or cultures (PDCs) that have RNASEQ and DNASEQ data available, allowing us to characterize all models by the CTX-S signature scores, CMS classes, and genotypes. We analyzed 477 CMS1-4 PDMR CRC models to further validate the findings from Moffitt and TCGA CRC tumors. The CTX-S score comparison analysis among 6 MUT vs. WT A/P/K subgroups (Figure 7) shows that the AP tumors had significantly higher scores than the A or P and the WT AP subgroups (*p* < 0.0001), whereas the APK tumors had significantly higher scores than the AK or PK and the K subgroups (*p* < 0.0001). Notably, the AP and APK tumors had similarly high CTX scores, whereas the K and the WT AP tumors that were APC and TP53 double wild-type had the lowest scores. These data validate our previously reported observations with human tumors [28].

When these PDMR models were compared among four CMS classes, the CMS2 tumors had the highest CTX-S scores that were significantly higher that all other three CMS classes (*p* < 0.0001), whereas the CMS4 tumors had the second highest scores that were higher than the CMS1 and CMS3 tumors (*p* < 0.0001) (Figure 8A). Notably, the AP and APK tumors appear much more frequently in CMS2 or CMS4 than CMS1 or CMS3 (Figure 8B). Moreover, the AP/APK tumors had significantly higher CTX-S scores than other nonAP/APK tumors across CMS classes (Figure 9). Similar patterns were also seen between the APK vs. nonAPK tumors in 213 MUT KRAS CMS1-4 models whereas a significant difference between AP vs. nonAP was only seen in CMS1 and CMS4 in 264 WT KRAS CMS1-4 models (see Appendix A).

### 3.7. The In Vitro Cetuximab Growth Inhibition Was Preferentially Associated with the CMS2 CRC Cell Lines Harboring MUT A/P Genotypes

A large number (*n* = 147) CRC cell lines with heterogeneous genetic backgrounds were analyzed for in vitro CTX sensitivity by Medico et al. [34]. We recently used these cell line data to validate the CTX-S signature score [28]. Here, we reanalyzed 91 of these cell lines (that had the data of APC, TP53, KRAS/NRAS, BRAF mutations) in association with the in vitro cetuximab-mediated growth inhibition data. The 91 cell lines were divided into six mutation subgroups including WT RAS/RAF cell lines: (i) AP; (ii) A or P; (iii) WT AP, and MUT RAS/RAF cell lines: (iv) APK; (v) AK or PK; (vi) MUT KB_others (all other KRAS/NRAS or BRAF-mutated cell lines). The waterfall plot of in vitro cetuximab growth inhibition (AUC index) vs. six mutation subgroups is shown in Figure 10. It shows that the majority of AP cell lines (10 out 15, 67%) and A/P cell lines (5 of 8, 62%) appear to be sensitive to CTX. Note that a few APK (3 of 21) and AK/PK (4 out 22) show some sensitivity to CTX while all KRAS-mutated cell lines harboring WT AP or BRAF (V600E) were not sensitive to CTX (Figure 10). These data indicate that APC and/or TP53 mutations were associated with in vitro cetuximab sensitivity.

Among 91 cell lines, 67 were classified as CMS1-4 subtypes: CMS1 (*n* = 13); CMS2 (*n* = 21); CMS3 (*n* = 12); CMS1 (*n* = 21). The CTX-S score comparison analysis among these CMS classes (Figure 11) shows that the CMS2 cell lines had significantly higher scores than all other CMS classes. Notably, the CMS2 cell lines appear to be more preferentially associated with the cell lines harboring the AP and A or P genotypes with higher AUC index values (S (AUC > 4000), Figure 11), suggesting that mutations in APC and/or TP53 may contribute to higher in vitro CTX sensitivity in the CMS2 cell lines.

## 4. Discussion

Recent studies suggest that CMS classification has a predictive role to guide the selection of anti-Vascular Endothelial Growth Factor (VEGF) therapy and EGFR inhibitor therapy [8,9,14,18]. However, the results have not always been consistent. The CALGB 80405 study suggested that the CMS2 class had better outcomes with EGFR inhibitor therapy while CMS1 class derived the best outcomes from anti-VEGF therapy [8]. However, the FIRE-3 study suggested no significant differences in survival in the CMS1 class, but the CMS4 class had significantly longer survival with EGFR inhibitor therapy [9]. Interestingly, the overall survival benefit of cetuximab in the CMS2 class seen in the CALGB 80405 study was not seen in the FIRE-3 study. Similarly, Okita et al. showed the association of CMS2 with improved progression-free survival and overall survival with irinotecan-based chemotherapy [16], and Mooi et al. noted that the CMS2 class had improved progression-free survival with bevacizumab [14]. In contrast, CMS2/CMS3 cohorts had better overall survival with EGFR inhibitor therapy in the CAIRO2 study [20]. These differences in the predictive role of CMS classification were attributed to different profiling strategies, tumor heterogeneity, and differences in the backbone chemotherapy [12,37,38,39]. It is not known if the differences in the prediction of benefit from EGFR inhibitor therapy in these studies could be related to the unique genomic mutational enrichment across the CMS classification.

We recently identified a 2-gene mutation signature of combined *APC* and *TP53* mutations in identifying cetuximab-sensitive CRC subpopulations [28]. This study aimed to use this 2-gene mutation signature to refine the CMS classification to better predict responses to cetuximab. The cetuximab sensitivity (CTX-S) signature scores of AP vs. non-AP tumors were determined across each of the CMS classes in Moffitt 433 CRC tumors. We found that AP mutated CRCs had significantly higher CTX-S scores compared to non-AP CRCs across all CMS classes, especially in CMS2 and CMS4 (both *p* < 0.0001). Tumors harboring combined AP mutations were predominantly enriched in the CMS2 class, and to a lesser degree, in the CMS4 class. Similar results were also obtained in independent CRC datasets of TCGA tumors (*n* = 531) and PDMR PDX/PDO/PDC models (*n* = 477). These results may help explain the clinical trial observations that CMS2 or CMS4 classes had better outcomes with EGFR inhibitor therapy in two separate Phase III studies [8,9]. Collectively, our data suggest that combination of the CMS classification with the 2-gene mutation signature (AP) has the potential to better predict cetuximab-sensitive CRC subpopulations. Results of this hybrid mutation/gene expression analysis also support the notion that in the future, integration of genomic markers with transcriptomic, stromal, and immune signatures will help develop and define better biomarker-driven treatment options for mCRC patients. In our study, despite the predictive role of this 2-gene mutation signature, we noted no clear prognostic role of the 2-gene AP mutation signature in Moffitt CRC tumors. This supports the concept that ideal biomarkers portend predictive but not prognostic effects. Notably, while we observed no statistically significance difference (trend *p* = 0.096) among CMS1-4 classes in Moffitt CRC, the CMS1 tumors appeared to have worse overall survival. Prior studies suggest CMS1 class to be least responsive to EGFR inhibitor therapy, especially among the metastatic tumors where it was noted to have the worst prognosis [8].

The CMS2 subtype was seen commonly in left sided tumors while CMS1 subtype were mostly right sided tumors [10]. These results were consistent with prior studies suggesting as we move from proximal to distal colon tumors, the prevalence of CMS1 decreases while the prevalence of CMS2 rises. Furthermore, our previous study noted that *APC* + *TP53* mutations are more common in left sided tumors than right sided tumors [28]. The right sided tumors are predicted to be resistant to EGFRi, whereas patients with left-sided tumors which more commonly harbor *APC* + *TP53* mutations are more likely responsive [40,41,42,43]. Notably we believe it is the combination of *APC* + *TP53* mutations that was consistently and significantly associated with the sidedness and EGFRi sensitivity [28].

Intriguingly, although RAS mutant tumors are known to be resistant to EGFR inhibitor therapy, our findings suggest that RAS mutated tumors, when co-existing with *APC* and *TP53* mutations (i.e., APK triple mutated tumors), had high CTX-S scores across all CMS classes with the highest sensitivity noted in the CMS2 subtype. These finding have led to the development of an ongoing clinical trial examining the potential for APK tumors to respond to cetuximab (NCT04853043). Biologically, our analyses suggest a hypothesis by which mutations in *APC + TP53* might enable WNT and p53 pathway “crosstalk” to transactivate the EGFR pathway, essentially addicting tumors to EGF ligands. Activation of the EGFR-PI3K-AKT signaling pathway has been clearly demonstrated in the *APC*^Min/+^ mouse by a mechanism involving upregulation of PGE2 [44,45]. Similar to WNT, the p53 pathway has crosstalk with the EGFR pathway. Specifically, mutant *TP53* has been shown to induce ERG1 transcription that is driven by p-ERK [46,47]. We found that the *APC + TP53* double-mutated tumors were predominantly the CMS2 subtype that was associated with WNT and MYC activation and frequent mutations in either *APC* or *TP53* [10,28]. Alternatively, mutations in *APC + TP53* may possibly enhance cetuximab sensitivity in wild-type and mutant *KRAS* CRCs via the mechanism involving antibody-dependent cellular cytotoxicity (ADCC) and/or immunogenic cell death (ICD). In addition to its inhibitory role in EGFR signaling, cetuximab, a human-murine chimeric IgG1 monoclonal antibody to EGFR, has been reported to induce ADCC and other immunogenic activities against the IgG1 mAb [48]. Cetuximab-mediated ADCC is mediated by natural killer (NK) and other immune cells such as dendritic cells (DCs), cytotoxic T cells, and macrophages [48,49,50,51,52,53,54,55,56,57,58,59,60]. Notably, some studies reported that cetuximab could induce ADCC in CRC cells regardless of *KRAS* mutation status [48,49,50]. Furthermore, cetuximab, in combination with chemotherapy, was also reported to induce ICD with an increase phagocytosis by dendritic cells (DCs), regardless of *KRAS* mutation status [61].

Notably, the RAS mutated tumors when associated with either *APC* or *TP53* mutations *alone* remain resistant to EGFR inhibitor therapy across all the CMS subtypes. Therefore, the 2-gene AP combinatorial mutation signature predicts responses to EGFR inhibitor therapy rather than individual mutations of *APC* or *TP53* or *KRAS.* In addition, we noted that the in vitro cetuximab growth inhibition was also preferentially associated with CRC cell lines harboring *APC* and/or *TP53* mutations, especially in the CMS2 cell lines, whereas all *KRAS*-mutated cell lines harboring *APC* and *TP53* double-wild type were not sensitive to CTX. These in vitro data suggest that mutations in either *APC* or *TP53* alone might sensitize some CRC cell lines to CTX. The difference between tumor and cell line data may be possibly due to the fact that the in vitro cell growth inhibition (in 2-D culture for a few days) may only partially reflect the in vivo cetuximab response. For example, stable disease (SD, a component of disease control response) that has been reported to achieved in a substantial percentage of cetuximab-treated *KRAS*-mutated CRC patients [62,63,64,65,66,67,68,69], can be only assessed during a long-term in vivo cetuximab treatment. The in vitro cetuximab-mediated cell growth inhibition study also did not investigate a potential role of ADCC and ICD.

The limitations of our study are the retrospective nature of analysis with the use of CTX-S score as a surrogate for cetuximab sensitivity due to the paucity of available data from patients treated with EGFR inhibitor therapy. The sample sizes, especially in CMS1 and CMS3 cohorts, are relatively small, making it difficult to make statistically meaningful comparisons for predicted cetuximab sensitivity between MUT A + P tumors vs. nonA + P tumors in these subclasses.

Several prognostic and predictive markers including MSI, RAS, *BRAF*, *HER2* mutational status are used in routine practice to guide systemic treatment options. Although currently RAS mutational status is considered predictive of EGFR inhibitor therapy, our study suggests a subset of RAS wild type—and even mutant tumors—may respond to EGFR inhibitor therapy. Due to notable toxicities associated with this class of therapeutics, it is prudent to select the patients most likely to respond to EGFR inhibitor therapy. Our study suggests that in addition to CMS2 classification, when carefully selected for 2-gene AP mutational signature, we can potentially identify additional patients within each CMS subclass that might respond to EGFR inhibitor therapy. Thus, cetuximab utility may not need to be restricted to just one CMS class.

Ultimately, CMS classification is a dynamic process with intra-tumoral variability and change in classification noted across the spectrum of premalignant to early-stage to advanced-stage tumors [39,70]. Future serial CtDNA assays now being evaluated in clinical trials may help us better understand the molecular heterogeneity and mechanisms of innate and acquired resistance pathways of EGFR inhibitor therapy to guide new drug development strategies.

## 5. Conclusions

The 2-gene mutation signature of combined *APC* and *TP53* (AP) mutations predicts responses to EGFR inhibitor therapy across all CMS classes, especially in CMS2 and CMS4. The CMS2 and/or CMS4 classes harbored the high frequency of AP mutations associated with high cetuximab sensitivity scores. Importantly, the CMS2 class was noted to have high CTX-S scores irrespective of AP mutational status suggesting that both CMS2 subtype and 2-gene AP mutation signature are independent biomarkers for the selection of patients for EGFR inhibitor therapy. The 2-gene AP mutational signature may thus help refine the CMS classification and identify a subset of patients with and without RAS mutations that derive maximal benefit from EGFR inhibitor therapy.

## Figures and Tables

**Figure 1 cancers-13-05394-f001:**
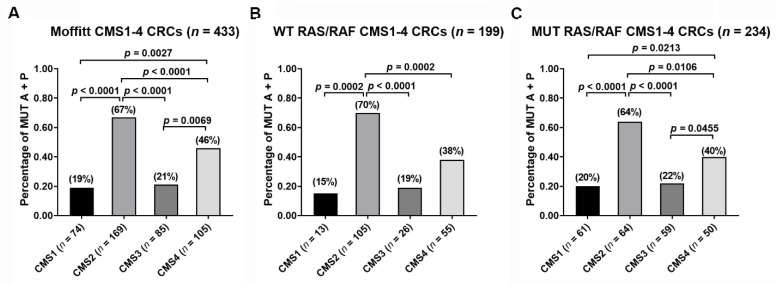
Comparison of the percentages of combined *APC* + *TP53* mutations (MUT A + P) across CMS1-4 classes. (**A**) All 433 CMS1-4 CRC tumors; (**B**) 199 CMS1-4 CRCs with WT RAS/RAF; (**C**) 234 CMS1-4 CRCs with MUT RAS/RAF. Percentages of MUT A + P in each of CMS1-4 classes are indicated. *p* values are for two-tailed Welch *t* test.

**Figure 2 cancers-13-05394-f002:**
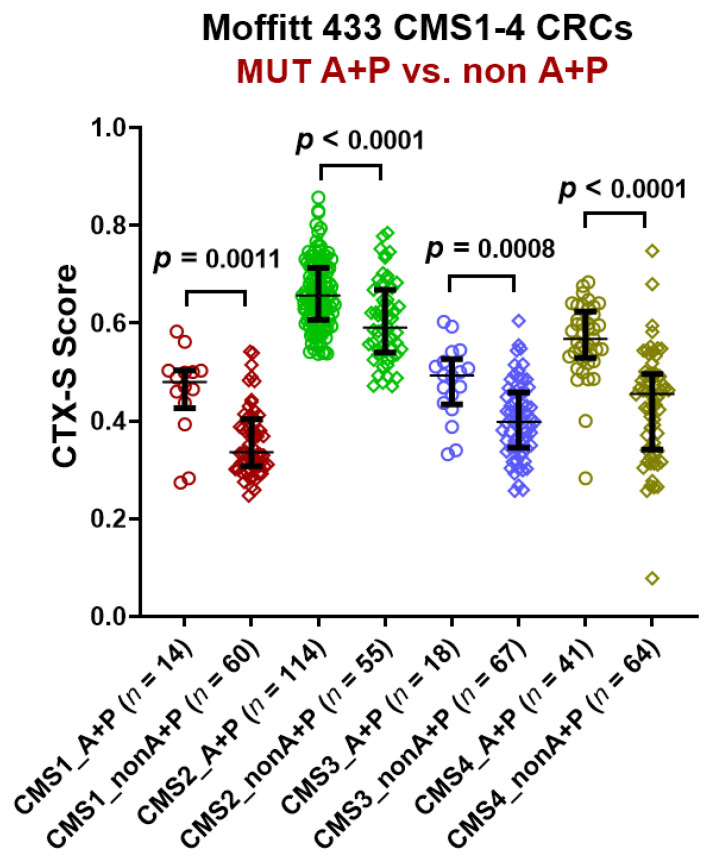
Comparison of CTX-S scores in MUT A + P vs. nonA + P in each of four CMS classes in Moffitt 433 CMS1-4 CRC tumors. Here, A—MUT *APC*; P—MUT *TP53;* MUT A + P represents tumors harboring combined *APC* + *TP53* mutations regardless of RAS/RAF mutation status. nonA + P represents tumors without combined *APC* and *TP53* mutations. Bars represent median with interquartile range. *p* values are for two-tailed Welch *t* test.

**Figure 3 cancers-13-05394-f003:**
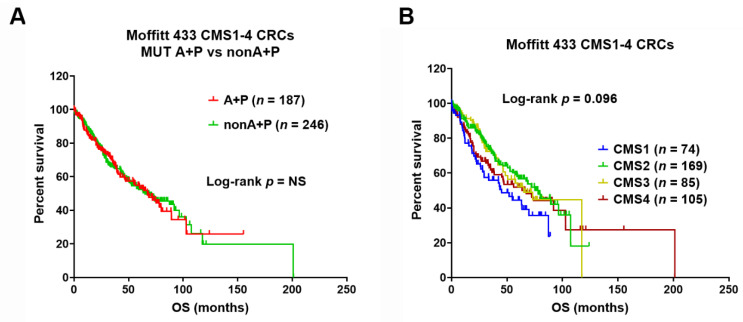
(**A**) Kaplan–Meier (KM) overall survival (OS) analysis by MUT *APC* + *TP53* (A + P) (*n* = 187) vs. non-A + P (*n* = 246) in Moffitt 433 CMS1-4 CRC tumors. (**B**) KM OS analysis across four CMS classes in 433 CMS1-4 CRCs.

**Figure 4 cancers-13-05394-f004:**
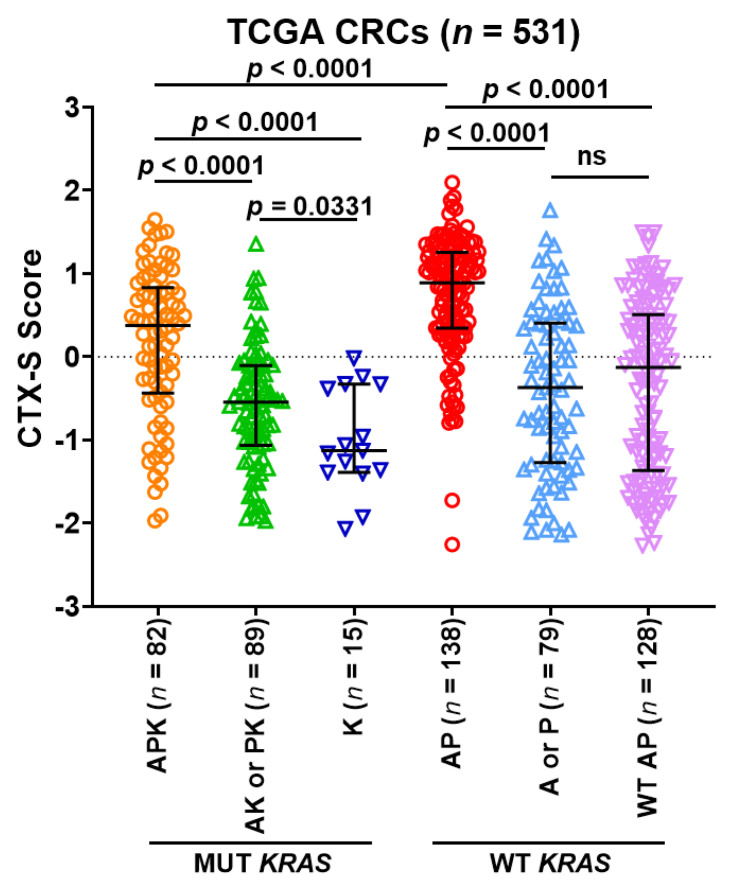
Comparison of CTX-S scores among six subgroups of MUT vs. WT *APC/TP53/KRAS* in TGGA CRC tumors. Here, A—Mut *APC*; P—Mut *TP53*; K—Mut *KRAS*; WT AP—both *APC* and *TP53* wild-type. Bars represent median with interquartile range. *p* values are for two-tailed Welch *t* test.

**Figure 5 cancers-13-05394-f005:**
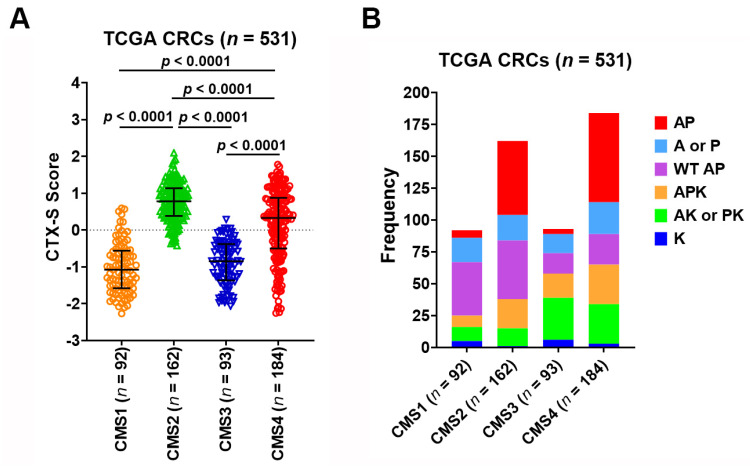
(**A**) Comparison of CTX-S scores among four CMS classes in TGCA CRC tumors. Bars represent median with interquartile range. *p* values are for two-tailed Welch *t* test. (**B**) Frequencies of six subgroups of MUT *APC* (A)/MUT *TP53* (P)/MUT *KRAS* (K) across CMS1-4 classes.

**Figure 6 cancers-13-05394-f006:**
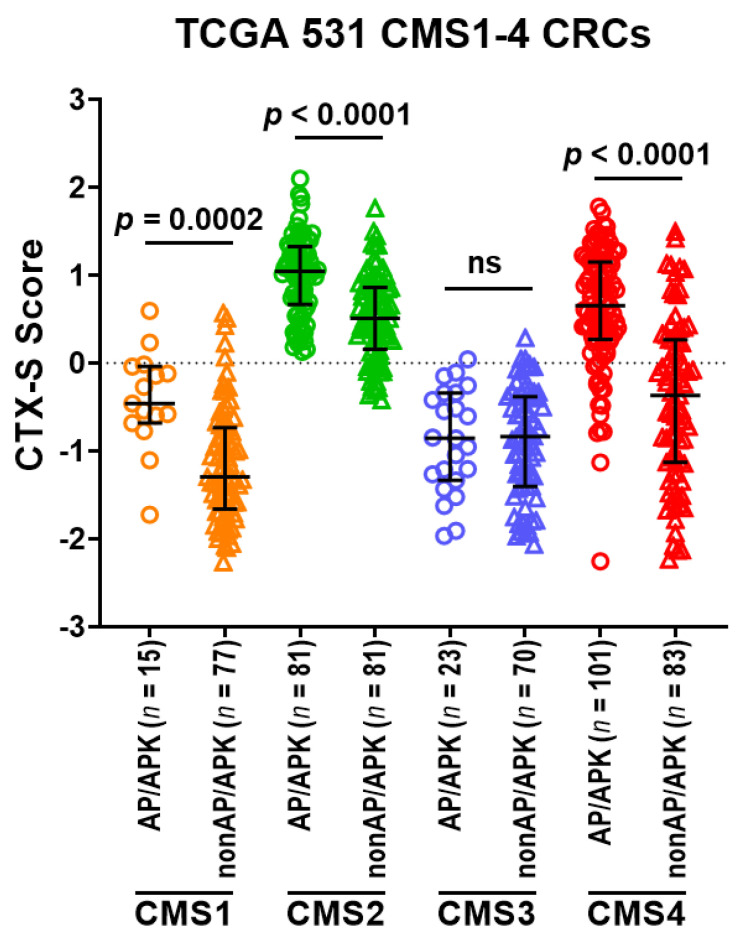
Comparison of CTX-S scores in AP/APK vs. nonAP/APK in each of four CMS classes in TCGA CRC tumors. Here, A—MUT *APC*, P—MUT *TP53*; K—MUT *KRAS*; AP/APK represents tumors harboring either AP or APK mutations; nonAP/APK represents tumors harboring neither AP nor APK mutations. Bars represent median with interquartile range. *p* values are for two-tailed Welch *t* test.

**Figure 7 cancers-13-05394-f007:**
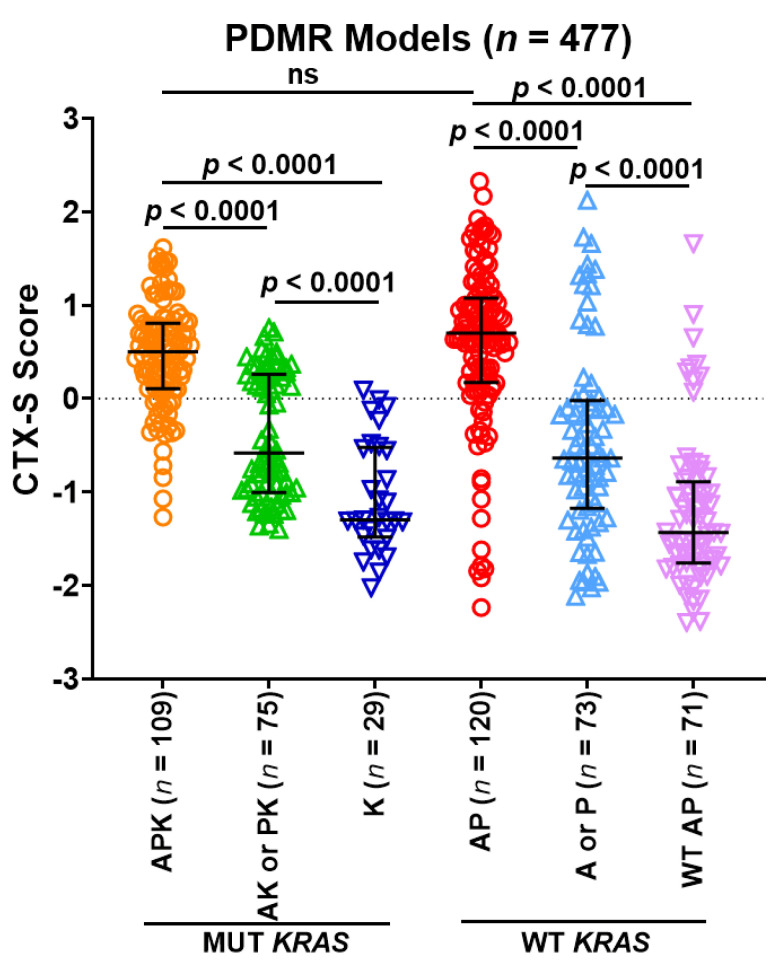
Comparison of CTX-S scores among six subgroups of MUT *APC* (A)/MUT *TP53* (P)/MUT *KRAS* (K) in PDMR CRC models. Bars represent median with interquartile range. *p* values are for two-tailed Welch *t* test.

**Figure 8 cancers-13-05394-f008:**
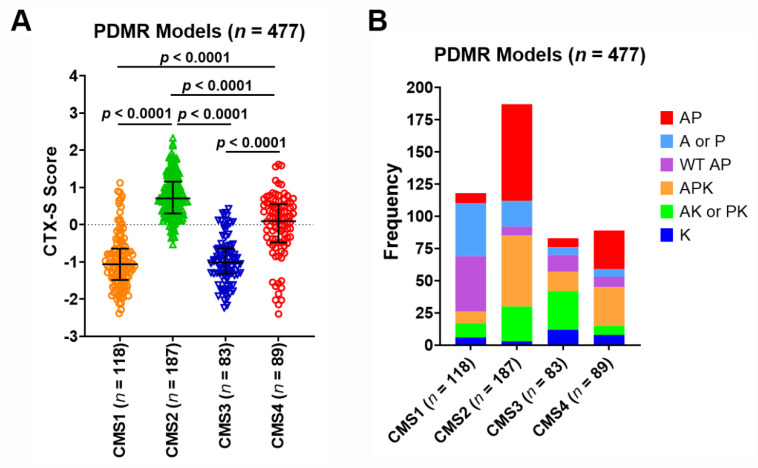
(**A**) Comparison of CTX-S scores among four CMS classes in PDMR CRC models. Bars represent median with interquartile range. *p* values are for two-tailed Welch *t* test. (**B**) Frequencies of six subgroups of MUT APC (A)/MUT TP53 (P)/MUT KRAS (K) across CMS1-4 classes.

**Figure 9 cancers-13-05394-f009:**
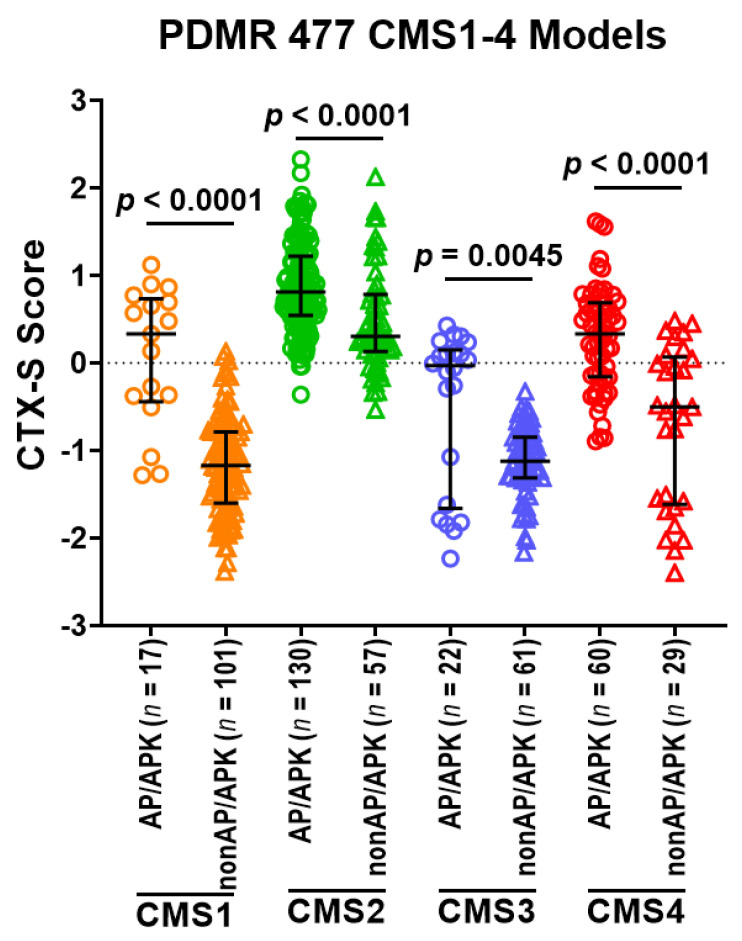
Comparison of CTX-S scores in AP/APK vs. nonAP/APK in each of four CMS classes in PDMR CRC models. Bars represent median with interquartile range. *p* values are for two-tailed Welch *t* test.

**Figure 10 cancers-13-05394-f010:**
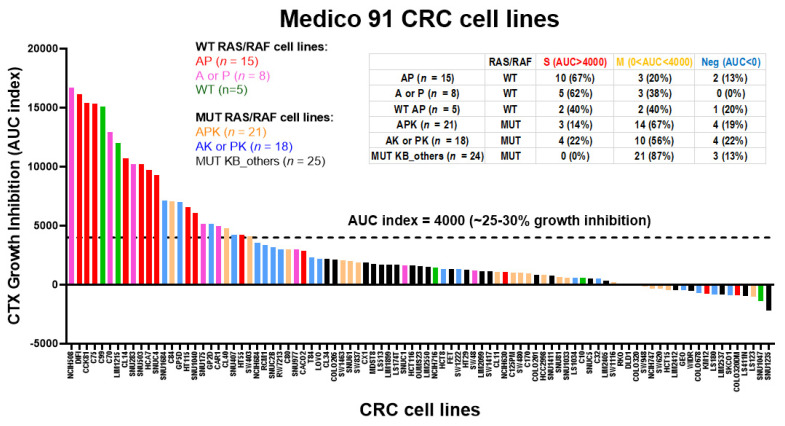
A diagram of in vitro growth inhibition by cetuximab (CTX) (AUC index) (from high to low) vs. 6 MUT vs. WT *APC/TP53/RAS(KRAS/NRAS)/BRAF* subgroups in Medico CRC cell lines. Notably, cell growth inhibition by cetuximab was measured by AUC-index representing the area under the concentration–inhibition curve of cetuximab at 0, 0.001, 0.01, 0.1, 1, 10, 25, 50, and 100 µg/mL, which measures overall dose-dependent drug inhibitory effects (Medico et al., 2015, ref [30]). That is, higher AUC index, higher growth inhibition. The AUC index = 4000 line represents up to ~25–30% of growth inhibition at higher CTX concentrations. Accordingly, S (sensitive) (AUC > 4000) represents >25–30% of growth inhibition; M (modest) (0 < AUC < 4000) represents 0 to 25–30% of growth inhibition; Neg (AUC < 0) represents negative effects on growth inhibition (i.e., CTX-treated cells had more growth than non-treated cells).

**Figure 11 cancers-13-05394-f011:**
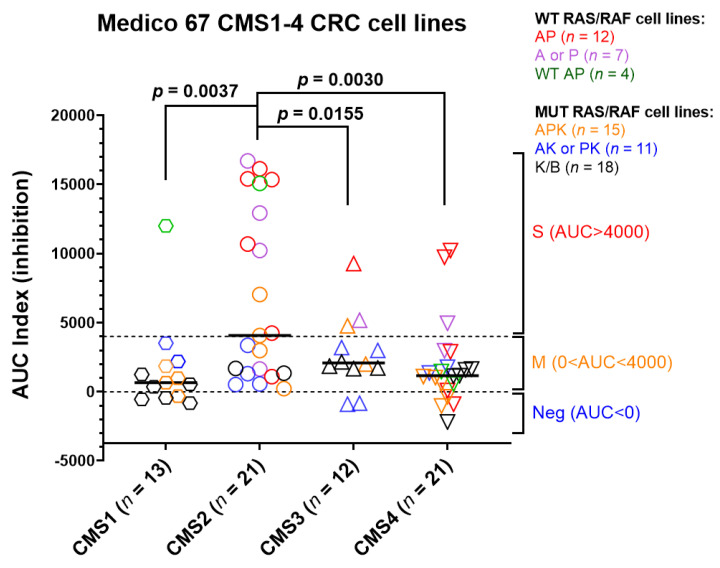
Comparison of CTX-S scores among four CMS classes in Medico CRC cell lines. Bars represent median with interquartile range. *p* values are for two-tailed Welch *t* test. Note that among 147 Medico cell lines treated with cetuximab, 67 cell lines are CMS1-4 classes with the mutation data of *APC, TP53, KRAS, NRAS,* and *BRAF*. The six MUT vs. WT A/P/K/N/B subgroups are highlighted by colors.

**Table 1 cancers-13-05394-t001:** Baseline characteristics of Moffitt 468 CRC tumors.

Characteristic	Total (*n* = 433)	CMS1 (*n* = 74)	CMS2 (*n* = 169)	CMS3 (*n* = 85)	CMS4 (*n* = 105)	*p*-Value
Age (median)	Mean (SD)	71.5 (11.6)	62.9 (11.9)	66.5 (13.6)	64.5 (13.0)	<0.001 ^a^
	Median (IQR)	74.0 (64.0, 80.8)	63.0 (55.0, 71.0)	69.0 (58.0, 78.0)	65.0 (55.8, 73.0)	-
	Range	(44.0, 93.0)	(34.0, 93.0)	(34.0, 90.0)	(30.0, 89.0)	-
Stage at diagnosis						
1	60 (14%)	9 (12.2%)	29 (17.2%)	15 (17.6%)	7 (6.7%)	0.016 ^b^
2	125 (29%)	28 (37.8%)	48 (28.4%)	23 (27.0%)	26 (24.8%)	-
3	141 (33%)	21 (28.4%)	49 (29.0%)	34 (40.0%)	37 (35.2%)	-
4	100 (23%)	15 (20.3%)	42 (24.8%)	10 (11.8%)	33 (31.4%)	-
Unknown	7 (1%)	1 (1.3%)	1 (0.6%)	3 (3.6%)	2 (1.9%)	
Specimen type						
Metastatic	90 (21%)	9 (12.2%)	39 (23.1%)	3 (3.5%)	41 (39%)	<0.001 ^b^
Primary	341 (79%)	65 (87.8%)	130 (76.9%)	82 (96.5%)	64 (61%)	-
Primary tumor						
Left	230 (53%)	24 (32.4%)	132 (79.5%)	42 (49.4%)	63 (60.6%)	<0.001 ^b^
Right	203 (47%)	50(67.6%)	34 (20.5%)	43 (50.6%)	41 (39.4%)	-
*APC* mutation ^c^						
Absent	146 (34%)	52 (70.3%)	17 (10.1%)	32 (37.6%)	45 (42.9%)	<0.001 ^b^
Present	287 (66%)	22 (29.7%)	152 (89.9%)	53 (62.4%)	60 (57.1%)	-
*TP53* mutations						
Absent	177 (41%)	34 (45.9%)	38 (22.5%)	57 (67.1%)	48 (45.7%)	<0.001 ^b^
Present	256 (59%)	40 (54.1%)	131 (77.5%)	28 (32.9%)	57 (54.3%)	-
RAS mutation						
Absent	248 (57%)	56 (75.7%)	113 (66.9%)	33 (38.8%)	59 (56.2%)	<0.001 ^b^
Present	185 (43%)	18 (24.3%)	56 (33.1%)	52 (61.2%)	46 (43.8%)	-
*BRAF* mutation ^d^						
Absent	381 (88%)	33 (44.6%)	169 (100%)	78 (91.8%)	101 (96.2%)	<0.001 ^b^
Present	52 (12%)	41 (55.4%)	0 (0%)	7 (8.2%)	4 (3.8%)	-
Microsatellite status						
Low	375 (87%)	32 (43.2%)	168 (99.4%)	72 (84.7%)	103 (98.1%)	<0.001 ^b^
High	58 (13%)	42 (56.8%)	1 (0.6%)	13 (15.3%)	2 (1.9%)	-

^a^ ANOVA; ^b^ Chi-squared test; ^c^ APC-truncating mutations; ^d^
*BRAF* (V600E).

**Table 2 cancers-13-05394-t002:** Frequency of stage-wise combined AP mutations (RAS/BRAF Wild Type) across the CMS classification in Moffitt CRCs.

Stage	*n*	CMS1	CMS2	CMS3	CMS4
		*n* (%)	*n* (%)	*n* (%)	*n* (%)
Total	104	2 (2)	77 (74)	5 (5)	20 (19)
Stage 1	14	0	13 (93)	1 (7)	0
Stage 2	27	0	21 (78)	0	6 (22)
Stage 3	36	1 (3)	25 (69)	4 (11)	6 (17)
Stage 4	27	1 (4)	18 (66)	0	8 (30)

## Data Availability

Not applicable.

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
