# Peer review of "APC and TP53 Mutations Predict Cetuximab Sensitivity across Consensus Molecular Subtypes"

_cancers, 2021, doi:10.3390/cancers13215394_

Round 1

Reviewer 1 Report

The study focused on APC and TP53 mutations predict cetuximab sensitivity across consensus molecular subtypes in colorectal cancer patients .

Usually, the treatment plan for cancer patients is established based on the TNM stage based on the size of the tumor and the presence of lymph nodes and distant metastases. However, it is known that the prognosis of colorectal cancer is not clearly predicted by TNM. The CMS classification method in colorectal caner tissue has been validated through several studies, and results that can confirm the response to treatment as well as the 5-year survival rate according to the type of tumor are being published. In this regard, this manuscript provides some information, but there is a need for clear data demonstrating the efficacy of cetuximab in patients with APC and TP53 mutations.

In this manuscript, a total of 1,055 CRC samples such as TCGA, NCI PDMR, and Medico were used as validation samples.

In the case of public data, quality control (QC) must be preceded, but the QC workflow is not specified. A specific reference to the QC workflow is required.

In addition, it was used dependently on the existing method without any added point to the existing method, "CMScaller(https://www.nature.com/articles/s41598-017-16747-x)“. The limitations of the lack of reliability of all sample data used and the reuse of existing methods without extension of characteristics are confirmed.

And CMS2 and CMS4 show the highest CTX-S, but first of all, it is difficult to see the difference between CMS1 and CMS3 clusters.

Finally overall survival data between the group with APC and TP53 mutations and the group without mutations, and overall survival data by CMS classification are presented. However, data on overall survival by cetuximab are not presented.

Therefore, there should be data on overall survival by cetuximab in the group of patients with APC and TP53 mutations according to CMS classification.

Author Response

Response to Reviewer 1:

We are grateful to the reviewer for thoughtful comments and suggestions. A point-by-point response is given below:

The study focused on APC and TP53 mutations predict cetuximab sensitivity across consensus molecular subtypes in colorectal cancer patients.

Usually, the treatment plan for cancer patients is established based on the TNM stage based on the size of the tumor and the presence of lymph nodes and distant metastases. However, it is known that the prognosis of colorectal cancer is not clearly predicted by TNM. The CMS classification method in colorectal cancer tissue has been validated through several studies, and results that can confirm the response to treatment as well as the 5-year survival rate according to the type of tumor are being published. In this regard, this manuscript provides some information, but there is a need for clear data demonstrating the efficacy of cetuximab in patients with APC and TP53 mutations.

In this manuscript, a total of 1,055 CRC samples such as TCGA, NCI PDMR, and Medico were used as validation samples. In the case of public data, quality control (QC) must be preceded, but the QC workflow is not specified. A specific reference to the QC workflow is required.

In addition, it was used dependently on the existing method without any added point to the existing method, "CMScaller (https://www.nature.com/articles/s41598-017-16747-x)“. The limitations of the lack of reliability of all sample data used and the reuse of existing methods without extension of characteristics are confirmed.

We appreciate the comment. In response to this comment, we have included additional technical information in Methods section and appropriate references were included regarding normalization/quality control method used. Moreover, we have generated new figures (Figure S1A, B, C) included in the supplementary materials to support the quality assessment of the data and the absence of significant outliers.

And CMS2 and CMS4 show the highest CTX-S, but first of all, it is difficult to see the difference between CMS1 and CMS3 clusters.

We appreciate the comment and want to acknowledge the limitations of sample size in CMS1 and CMS3 in few datasets. We revised the discussion to reflect such limitations.

Finally overall survival data between the group with APC and TP53 mutations and the group without mutations, and overall survival data by CMS classification are presented. However, data on overall survival by cetuximab are not presented. Therefore, there should be data on overall survival by cetuximab in the group of patients with APC and TP53 mutations according to CMS classification.

We would like to clarify that none of the patients included in the analyses were actually treated with cetuximab. As it turns out, there are very few trials that used cetuximab and stored tumor samples for future uses such as ours. Moreover, these samples are frequently exhausted when found.  Because of lack of access to patients and their tumor samples treated with cetuximab, we used previously clinically validated cetuximab-sensitivity score as a surrogate for sensitivity to cetuximab in Moffitt and other validation datasets. Therefore, unfortunately we cannot provide overall survival of patients treated with cetuximab. Notably, we have a prospective clinical trial going on to validate these results (NCT04853043).

Response to Reviewer 1:

We are grateful to the reviewer for thoughtful comments and suggestions. A point-by-point response is given below:

The study focused on APC and TP53 mutations predict cetuximab sensitivity across consensus molecular subtypes in colorectal cancer patients.

Usually, the treatment plan for cancer patients is established based on the TNM stage based on the size of the tumor and the presence of lymph nodes and distant metastases. However, it is known that the prognosis of colorectal cancer is not clearly predicted by TNM. The CMS classification method in colorectal cancer tissue has been validated through several studies, and results that can confirm the response to treatment as well as the 5-year survival rate according to the type of tumor are being published. In this regard, this manuscript provides some information, but there is a need for clear data demonstrating the efficacy of cetuximab in patients with APC and TP53 mutations.

In this manuscript, a total of 1,055 CRC samples such as TCGA, NCI PDMR, and Medico were used as validation samples. In the case of public data, quality control (QC) must be preceded, but the QC workflow is not specified. A specific reference to the QC workflow is required.

In addition, it was used dependently on the existing method without any added point to the existing method, "CMScaller (https://www.nature.com/articles/s41598-017-16747-x)“. The limitations of the lack of reliability of all sample data used and the reuse of existing methods without extension of characteristics are confirmed.

We appreciate the comment. In response to this comment, we have included additional technical information in Methods section and appropriate references were included regarding normalization/quality control method used. Moreover, we have generated new figures (Figure S1A, B, C) included in the supplementary materials to support the quality assessment of the data and the absence of significant outliers.

And CMS2 and CMS4 show the highest CTX-S, but first of all, it is difficult to see the difference between CMS1 and CMS3 clusters.

We appreciate the comment and want to acknowledge the limitations of sample size in CMS1 and CMS3 in few datasets. We revised the discussion to reflect such limitations.

Finally overall survival data between the group with APC and TP53 mutations and the group without mutations, and overall survival data by CMS classification are presented. However, data on overall survival by cetuximab are not presented. Therefore, there should be data on overall survival by cetuximab in the group of patients with APC and TP53 mutations according to CMS classification.

We would like to clarify that none of the patients included in the analyses were actually treated with cetuximab. As it turns out, there are very few trials that used cetuximab and stored tumor samples for future uses such as ours. Moreover, these samples are frequently exhausted when found.  Because of lack of access to patients and their tumor samples treated with cetuximab, we used previously clinically validated cetuximab-sensitivity score as a surrogate for sensitivity to cetuximab in Moffitt and other validation datasets. Therefore, unfortunately we cannot provide overall survival of patients treated with cetuximab. Notably, we have a prospective clinical trial going on to validate these results (NCT04853043).

Response to Reviewer 1:

We are grateful to the reviewer for thoughtful comments and suggestions. A point-by-point response is given below:

The study focused on APC and TP53 mutations predict cetuximab sensitivity across consensus molecular subtypes in colorectal cancer patients.

Usually, the treatment plan for cancer patients is established based on the TNM stage based on the size of the tumor and the presence of lymph nodes and distant metastases. However, it is known that the prognosis of colorectal cancer is not clearly predicted by TNM. The CMS classification method in colorectal cancer tissue has been validated through several studies, and results that can confirm the response to treatment as well as the 5-year survival rate according to the type of tumor are being published. In this regard, this manuscript provides some information, but there is a need for clear data demonstrating the efficacy of cetuximab in patients with APC and TP53 mutations.

In this manuscript, a total of 1,055 CRC samples such as TCGA, NCI PDMR, and Medico were used as validation samples. In the case of public data, quality control (QC) must be preceded, but the QC workflow is not specified. A specific reference to the QC workflow is required.

In addition, it was used dependently on the existing method without any added point to the existing method, "CMScaller (https://www.nature.com/articles/s41598-017-16747-x)“. The limitations of the lack of reliability of all sample data used and the reuse of existing methods without extension of characteristics are confirmed.

We appreciate the comment. In response to this comment, we have included additional technical information in Methods section and appropriate references were included regarding normalization/quality control method used. Moreover, we have generated new figures (Figure S1A, B, C) included in the supplementary materials to support the quality assessment of the data and the absence of significant outliers.

And CMS2 and CMS4 show the highest CTX-S, but first of all, it is difficult to see the difference between CMS1 and CMS3 clusters.

We appreciate the comment and want to acknowledge the limitations of sample size in CMS1 and CMS3 in few datasets. We revised the discussion to reflect such limitations.

Finally overall survival data between the group with APC and TP53 mutations and the group without mutations, and overall survival data by CMS classification are presented. However, data on overall survival by cetuximab are not presented. Therefore, there should be data on overall survival by cetuximab in the group of patients with APC and TP53 mutations according to CMS classification.

We would like to clarify that none of the patients included in the analyses were actually treated with cetuximab. As it turns out, there are very few trials that used cetuximab and stored tumor samples for future uses such as ours. Moreover, these samples are frequently exhausted when found.  Because of lack of access to patients and their tumor samples treated with cetuximab, we used previously clinically validated cetuximab-sensitivity score as a surrogate for sensitivity to cetuximab in Moffitt and other validation datasets. Therefore, unfortunately we cannot provide overall survival of patients treated with cetuximab. Notably, we have a prospective clinical trial going on to validate these results (NCT04853043).

Reviewer 2 Report

Thota and colleagues have used APC and TP53 mutation in consensus molecular subtype (CMS) classification used as a predictive model to identify cetuximab drug sensitivity in colorectal cancer. The authors used data from TCGA and PDMR of CRC patients and colon cancer cell lines. The current study design is interesting and scientifically sound. Though in table 1, the authors provided mutation distribution in each class, however, authors haven't provided information on whether CRC patients had received any therapy before resection of the tumor? 
Cetuximab inhibits EGFR by preventing binding to its ligand. I am wondering if the authors can provide any possible relation between EGFR ligand binding and APC+TP53 mutation and add the information of EGFR mutations from these data? 
In table 1, indicating the location of the primary tumor as left and right is a little confusing without appropriate discussion and without additional information which can provide its significance on the outcome? As mentioned in Line 65 - 68. authors might need to segregate the frequency of APC and TP53 mutation in left and right tumors.        

Author Response

Response to Reviewer 2:

We are grateful to the reviewer for thoughtful comments and suggestions. A point-by-point response is given below:

Thota and colleagues have used APC and TP53 mutation in consensus molecular subtype (CMS) classification used as a predictive model to identify cetuximab drug sensitivity in colorectal cancer. The authors used data from TCGA and PDMR of CRC patients and colon cancer cell lines. The current study design is interesting and scientifically sound. Though in table 1, the authors provided mutation distribution in each class, however, authors haven't provided information on whether CRC patients had received any therapy before resection of the tumor? 

We appreciate the comment. These are CRC patients whose samples were collected at the time of diagnosis. They didn’t receive any therapy prior to the tumor resection.

Cetuximab inhibits EGFR by preventing binding to its ligand. I am wondering if the authors can provide any possible relation between EGFR ligand binding and APC+TP53 mutation and add the information of EGFR mutations from these data? 

We appreciate this important biological question highlighting the EGFR ligand binding. In our dataset we do not have ability to confirm these results and we are not aware of any similar studies in CRC but through our recently funded R21 (R21CA25531201A1 Yeatman)         we are currently investigating if APC + TP53 mutations may enhance EGFR ligand binding to increase CRC addiction to EGFR signaling, resulting in increased sensitivity to an EGFR inhibitor.

Notably, our analyses suggest a hypothesis by which mutations in APC+TP53 might enable WNT and p53 pathway “crosstalk” to transactivate the EGFR pathway, essentially addicting tumors to EGF ligands. Activation of the EGFR-PI3K-AKT signaling pathway has been clearly demonstrated in the APCMin/+ mouse by a mechanism involving upregulation of PGE2 [44,45]. Similar to WNT, the p53 pathway has crosstalk with the EGFR pathway. Specifically, mutant TP53 has been shown to induce ERG1 transcription that is driven by p-ERK [46,47]. We found that the APC + TP53 double-mutated tumors were predominantly the CMS2 subtype that was associated with WNT and MYC activation and frequent mutations in either APC or TP53 [10,28]. We have included such discussion in our manuscript.In table 1, indicating the location of the primary tumor as left and right is a little confusing without appropriate discussion and without additional information which can provide its significance on the outcome? As mentioned in Line 65 - 68. authors might need to segregate the frequency of APC and TP53 mutation in left and right tumors.        

We appreciate the comment. As suggested, to highlight the significance of sidedness in CRC, we have included following statements in Results and Discussion:
